# Fatigue Failure in Engineered Components and How It Can Be Eliminated: Case Studies on the Influence of Bifilms

**John Campbell** [1] and **Murat Tiryakioğlu** [2,*]

1    School of Metallurgy and Materials, University of Birmingham, Birmingham B15 2TT, UK
2    School of Engineering and Technology, Jacksonville University, Jacksonville, FL 32211, USA
*    Correspondence: mtiryak@ju.edu

**Abstract:** The fatigue of engineered components involves more than the fatigue of metals as studied in laboratories. The miniscule laboratory test pieces cannot represent the pre-existing macroscopic crack defects in real engineering components. This brief study illustrates five examples in which major cracks are pre-existing as a result of the presence of bifilm defects. The pre-existing defects account for up to 90 per cent of the failure of so-called fatigue failure. The presence of pre-existing bifilm defects is of overwhelming importance. It is, with regret, suggested that the attempts at the elimination of so-called fatigue failures by only studying fatigue is misguided. The so-called fatigue failures of engineering components can be understood and addressed by realizing the major contributions of bifilms.

**Keywords:** entrainment; pores; oxide films; premature fracture





## 1. Introduction

Since the Versailles train accident, due to a broken axle in 1842, we have become aware of the failure of engineered components well below their yield strength [1]. This incident, which baffled all the scientists and engineers at the time, led to the emergence of the phrase "metal fatigue"; because there was no consensus in the scientific community about the reason for the premature and unexpected failure, it was decided that metal probably "got tired". In the last 180 years, we have become aware of the underlying reasons for fatigue failure, how fatigue cracks are initiated, how they propagate and when they lead to sudden final fracture. Especially in the last 50 years, the combination of metallurgical advances that address fatigue crack initiators, such as pores, and the emergence of fracture mechanics models, some of which incorporate statistical tools, has led to the accurate prediction of fatigue performance in laboratory settings, and, in a large majority of cases, in industrial applications. Yet, train axles are still reported [2] to fail, leading to catastrophic loss of life [3]. It is, in a sense, unspeakably bad taste to draw attention to the incomplete success of the science in the context of, for example, the dreadful horror experienced by those in a helicopter, subjected to so-called 'fatigue failure' of its main drive shaft, as it plunges to the ground. The authors deeply regret that at times it seems necessary to cite these horrors to alert us to the seriousness of the shortcomings of our science and technology. Our scientific predictions are, at times, clearly susceptible to calamitous and tragic failure. This paper is motivated by this gap.

## 2. Background

Fatigue failure is very common, but its control clearly is not yet sufficiently understood to enable engineering components to claim invulnerability. Otherwise, why would the bearings of wind turbines continue to fail by fatigue, despite the current efforts of mechanical engineers, fracture mechanics and metallurgical experts and manufacturers? Why would turbine blades fail in airborne aircraft? Why would critical parts of drive trains fail in airborne helicopters, even when parts are designed with safety factors of five?

These are disturbing facts, strongly suggesting that some other, as yet unknown and invisible factor, may be involved. It is proposed here that the additional factor is the presence of prior cracks in materials, as was suggested by Griffiths [4] a century ago. This is nothing new. What is new is the discovery of the nature of the cracks, their source, the astonishing density of some of their populations, and their astonishing range of sizes, from microns to fractions of metres in some cast products [5].

Considering the usual manufacturing route for metals, the metal is melted, alloyed, and cast into ingots of some kind, which are subsequently worked by forging, rolling or extrusion etc. Almost all our current casting processes involve pouring, in which the surface oxide film, a ceramic, is entrained and submerged, along with air, as depicted schematically in Figure 1. The entrainment mechanism results in the creation of double oxide films, i.e., bifilms [6], in the following processes: (i) folding over of the liquid surface, (ii) by impact between droplets, and/or (iii) entrainment of air, which subsequently reacts with the liquid metal to form more oxide films. All of these processes create double films with dry-side-to-dry-side contact so that no bonding can occur between these ceramic interfaces; they act as cracks in the liquid. An example is presented in Figure 2 [7]. Conversely, the outer faces of the bifilm are in perfect atomic contact with the matrix. The simultaneous properties of zero and total bonding within the bifilm are unique features of this defect. Hence, bifilms are cracks in metals. At this time, it is a significant misfortune that practically all our engineering metals needlessly contain dense populations of cracks introduced by our poor casting processes. The oxide-to-oxide interfaces ensure that such cracks remain stubbornly resistant to bonding, despite significant amounts of plastic working and even hot isostatic pressing [8,9].

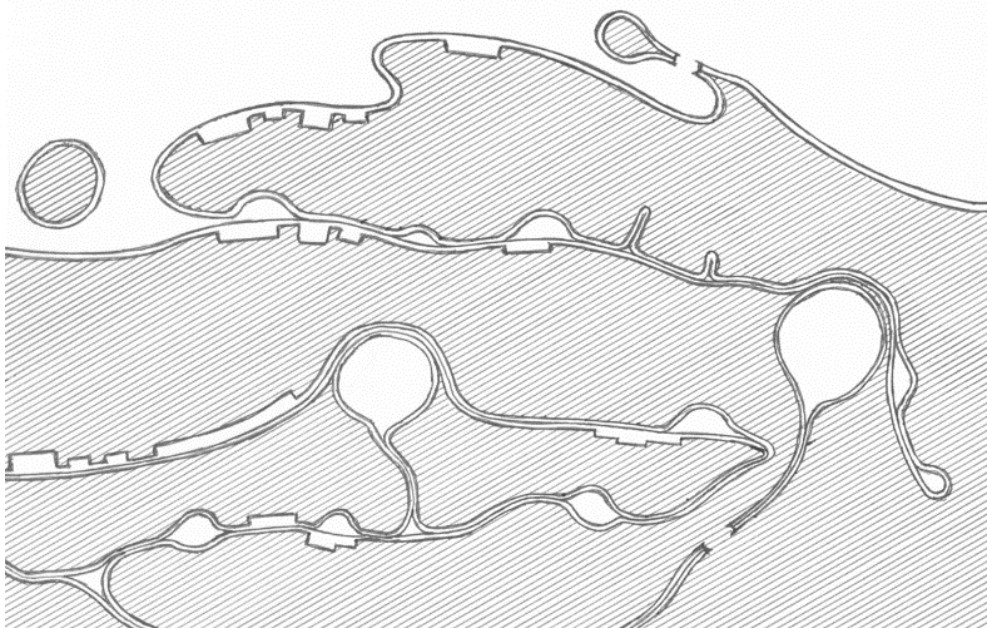

**Figure 1.** Turbulence in the liquid, causing entrainment of the surface oxide, creating bifilms and bubbles.

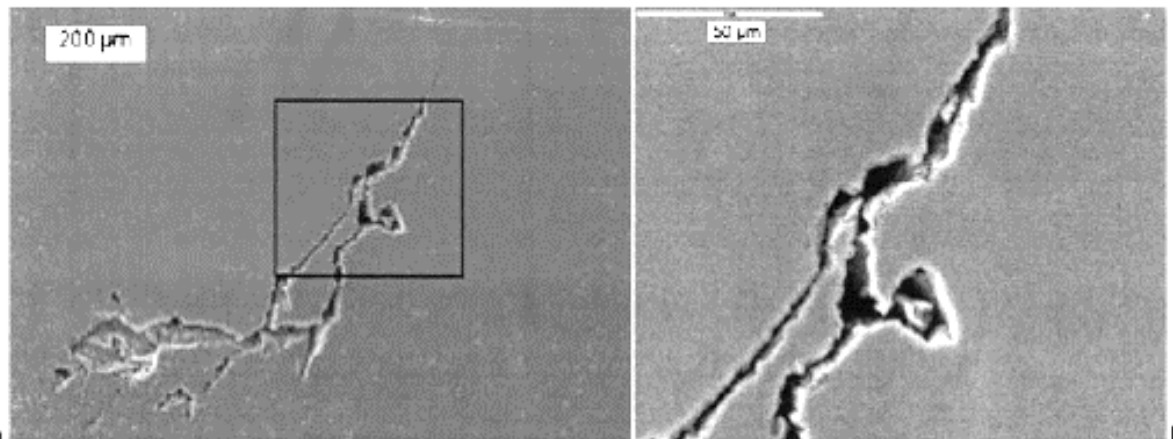

**Figure 2.** A folded-over surface oxide film (bifilm) on a polished surface in an Al-7%Si-Mg casting Reprinted with permission from ref. [7]. Copyright 2022 Elsevier. (**a**) the entrained films (**b**) the square in (**a**) magnified to show the double nature of the film.

## 3. Bifilms: The Pre-Existing Cracks in Our Engineered Components

Most of our casting processes entrain masses of bifilm cracks into the liquid metal. This damage is cumulative, i.e., the entrainment damage produced during ingot manufacturing is added to damage due to disruption of the metal surface, such as in degassing and, especially, during liquid metal transfers. When combined with new damage during mold filling, it is, in fact, amazing that our castings have been able to perform as structural components. This cumulative damage, of course, survives as a dense population of cracks during solidification.

Initially, the bifilms are scrambled by the powerful bulk turbulence of pouring, and so become compact, raveled structures which, because of their compact and convoluted form, are less harmful in terms of their effects on mechanical properties.

During solidification, however, the growth of dendrites or grains can push the bifilms, organizing them into flattened, parallel cracks which can have a major impact on properties (Figure 3). It is worth emphasizing that these cracks, which originate in the liquid state, do so by a stress-free mechanism.

Many grain boundaries of metals exhibit bifilms, as witnessed by their precipitates at these boundaries, whereas other boundaries remain clear. The bifilm reduces the strain energy of formation of the precipitate; its volume and shape changes, partly accommodated by the 'air gap' of the bifilm, greatly reducing the energy expended on local plastic deformation, compared to the interfacial energy savings generally assumed in traditional metallurgy, which are comparatively negligible [10]. In general, therefore, precipitates form on bifilms, not on grain boundaries.

Similar action to straighten bifilms can happen if gas in solution in the metal diffuses into the bifilm to inflate it partially, or as pressure is reduced by solidification shrinkage, as is clearly visible in the X-ray radiographs of Figure 4, comparing an as-cast sample of an Al alloy frozen in air, and an otherwise identical sample frozen under a reduced pressure [11]. Figure 4a shows the as-cast faint images of compact bifilms, temporarily raveled by the turbulence of pouring during the filling of the mould. In Figure 4b, those same bifilms are somewhat inflated and straightened, becoming clearly visible. Their number, size and density cause them to mutually impinge, generating an irregular lattice of cracks throughout the liquid.

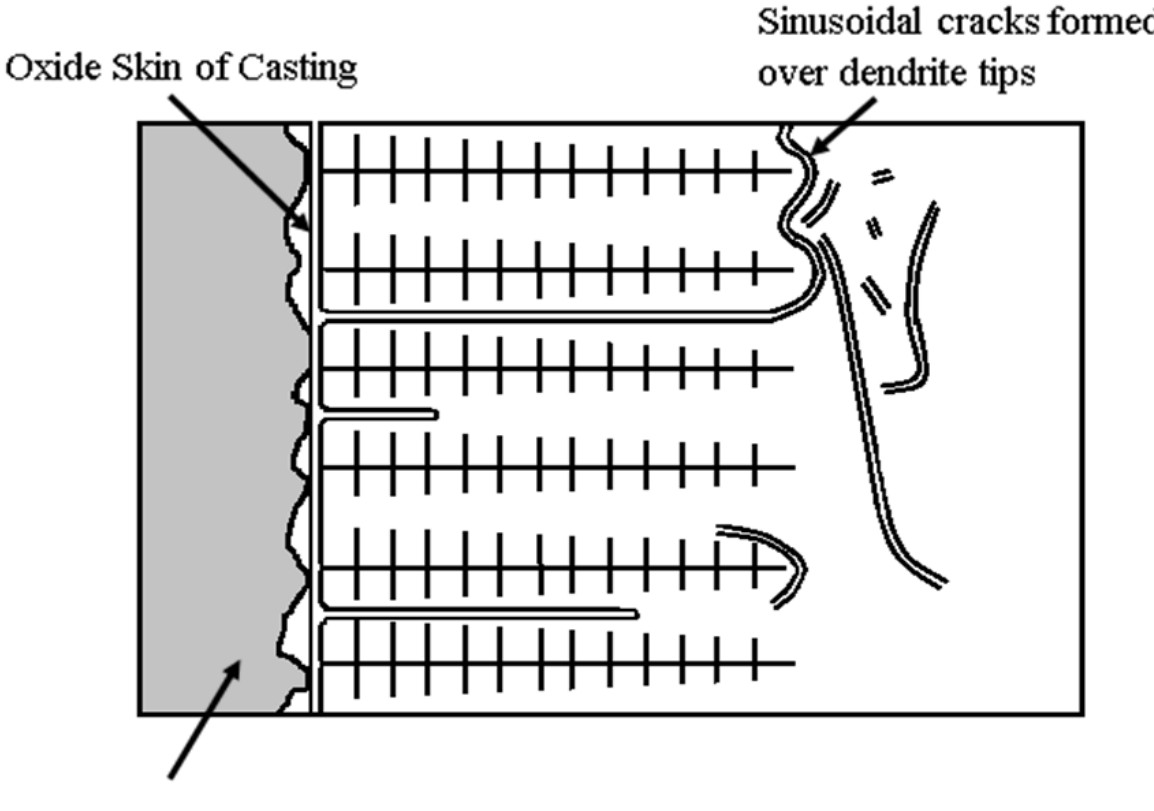

**Figure 3.** Bifilm cracks, floating randomly in the liquid metal, can be straightened (flattened) by dendritic growth, resulting in parallel cracks and sinusoidal cracks (the latter often being intergranular cracks).

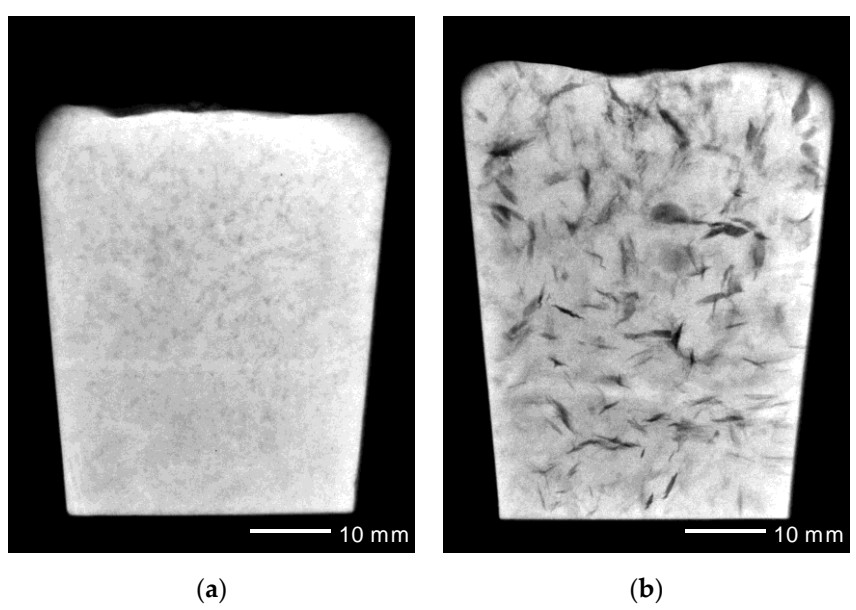

**Figure 4.** (**a**) Aluminium alloy frozen in air; (**b**) identical alloy frozen under 0.1 atm pressure Reprinted with permission from ref. [11]. Copyright 2022 Elsevier.

During solidification, the unbonded surfaces of the bifilms sometimes open up by solidification shrinkage and/or inward gas diffusion, forming pores in which dendrites are usually exposed. The only mechanism permitting the formation of these pores is

the prior existence of a bifilm, because pores cannot nucleate, either homogeneously or heterogeneously, in solidifying metals [12–19]. Pores are only the visible part of the damage suffered by the liquid metal, but most damage remains hidden even when pores are present in castings [20]. That is why attributing crack initiation to the presence of a pore is probably an error, as there are usually inactive bifilms around and extending from the pores, as was demonstrated by cracks appearing around pores early during in situ tensile tests [21] of 319 aluminium alloy castings. Yet, perversely, the main effort to improve the structural quality of castings has been given to minimization of pores. Processes such as squeeze casting can eliminate the pores but the damage remains, this time invisible, allowing the parts to go through all non-destructive tests without being detected. Similarly, hydrogen removal via degassing mostly hides the entrainment damage in castings by keeping bifilms uninflated (closed). Secondary processes, such as hot isostatic pressing (HIP), also closes pores, but bifilms stay almost completely resistant to HIP and continue to act as cracks, especially in aluminium castings [8,9]. Hence, the two primary densifying processes in Al alloy castings, hydrogen degassing and HIP, are both unhelpful in eliminating the root cause of the metal fatigue failure, i.e., the bifilms.

In carbon steels the situation is more complicated. The high temperatures and favorable chemistries sometimes cooperate to liquefy the surface oxide, so that bifilms cannot be formed (the folding or impact events between molten oxides are liquid-to-liquid, merely forming oxide droplets). Boron steels and high Mn steels are good examples displaying unusually high toughness and ductility as a result of their liquid oxides when in the liquid state, and consequent low bifilm content in the solidified and wrought product. In other steels, even if the oxide remains solid, the thin films tend to transform into particles by a coarsening reaction to reduce surface energy. In contrast, at the lower processing temperatures of many stainless steels and Ni-based alloys, bifilms tend to retain their original form, and are, therefore, numerous and dangerous. One of the authors (JC) has memories of trying, without much success, to find evidence of any metal on fracture surfaces of some as-cast stainless and Ni alloys castings; the oxide bifilm (one half of the bifilm on each of the opposed fracture surfaces) coverage was close to 100 per cent, explaining the rather low properties.

It should be noted that manufacturing techniques have been recently developed which can contribute to the elimination of these defects, eliminating, in turn, the immediate initiation of a fatigue crack and inhibiting its propagation. However, this aspect cannot be covered in this short account. These issues are dealt with at length elsewhere [22].

## 4. Bifilms and Fatigue Crack Initiation and Propagation

In several studies, cracks were observed to grow from structural defects at, or shortly after, the first stress cycle [23–26]. In one of these studies, Wang et al. [26] observed that a crack opened up near a pore in the first tensile stress cycle in an A319 aluminium alloy specimen. It is the opinion of the authors that these cracks that form at, or near, the first cycle are not fatigue cracks, but bifilms that are near, or are part of, the pore. In the same alloy, cracks observed by Yousefian et al. [20] were identified as bifilms that remained hidden (invisible) near a pore, even though the sample was allowed to solidify in reduced pressure to expand the pores as much as possible. Another pore with a clear bifilm through its centre is provided by Le et al. [27] in their study on the high cycle fatigue of a cast aluminium alloy specimen. Le et al. [27] referred to this bifilm as a fissure, a defect commonly reported in aluminium products, such as in castings [28,29], as well as drawn aluminium alloy wire [30]. As can be expected, bifilms were observed [21,31] to open up during in situ tensile tests of cast aluminium alloy specimens as well. Hence, bifilms are the reason for the short, or nonexistent, crack initiation phases in cast alloys.

Recently, there have been several in situ fatigue studies in which the crack propagation in cast aluminium alloy specimens has been studied. Bogdanoff et al. [32,33] reported that secondary cracks opened up unexpectedly away from the notch that was machined into the specimens. Analysis of the secondary cracks observed in interrupted tests showed

that these cracks were bifilms. In another in situ study by X-ray tomography, many cracks were observed near, and distant from, the pores at the first cycle; the subsequent crack propagation was simply the coalescence of many cracks between pores. This explains why fracture surfaces of high cycle fatigue specimens of cast aluminium alloys have been reported [34] to have unexpectedly large areas of so-called cleavage, instead of typical striations. Fatigue crack growth, aided by the formation of secondary cracks, both around and away from the crack tip, has been reported and shown in many studies [35]. These secondary cracks were mistakenly attributed to "weak and brittle" Si particles and/or inter-metallics. Si particles and inter-metallics are not only very strong but also quite ductile [36]. Hence, they are not intrinsically weak, but extrinsically weakened by the bifilm on which they have precipitated during solidification. Moreover, it was found in an interrupted in situ fatigue test that many bifilms opened as cracks in front of the growing crack [33]. The finding that fatigue crack growth in B319 aluminium alloy specimens, involving multiple cracks growing and later coalescing [37], provides further evidence that fatigue crack growth is dominated by the degree of damage to the liquid metal. This damage from the casting process has been mistakenly interpreted as fatigue damage in the literature.

It is the authors' opinion that the success of fatigue studies on metals, especially cast alloys, remains incomplete, because they have yet to address the following issues:

- Fatigue crack initiation has to be attributed to the pre-existing cracks, bifilms, due to damage to the liquid metal. There is now growing evidence for this from in situ studies. Pores are only the visible parts of the damage, being the inflated regions of some bifilms. Pores assist their surrounding regions of bifilm, and nearby bifilms, to open up at low stresses by serving as stress concentrators.
- Fatigue crack propagation often appears to be dominated by secondary cracks and/or other cracks opening up and later coalescing. Hence, the fatigue crack propagation data contain the extrinsic effect of bifilms.
- The size of bifilms can be much larger than the fatigue specimens.

The authors are confident that these issues will be addressed as better understanding of the lasting effects of liquid metal damage is developed.

Let us now go through some case studies on the fatigue failure of some engineered components in service and discuss the role of bifilms in catastrophic fatigue failures. Such an analysis for a failed wind turbine bearing has been recently provided by one of the authors, discovering that the bifilm contributed over 90% of the failure, whereas fatigue contributed only approximately 1% [38].

## 5. Case Studies on In-Service Fatigue Failure Due to Bifilms

### 5.1. Case 1: Fatigue Failure of a Turbine Blade

Our first example is the catastrophic failure of a turbine blade which led, in turn, to the failure of many of the other blades of the turbine (Figure 5), causing a Cessna aircraft to crash with the loss of lives [39].

The fracture surface of the cast Ni alloy turbine blade which failed due to fatigue is shown in Figure 6. The fracture surface is somewhat rugged, tending to follow the grains of this polycrystalline casting. However, only a very few grains on the fracture surface showed striations. For much of the fracture surface the form of some of the 'grains' is curious, perhaps best described as quasi-cleavage (a name to conceal our ignorance of the fast-cracking mechanism, which is definitely not cleavage, but probably the opening of a bifilm). Other quasi-cleavage grains tended to resemble conchoidal fractures, which is a failure mode to be expected in a brittle glass, not a ductile metal. It is suggested that these features are bifilms, ballooning out like the sails of a square-rigged ship in the powerful flow of metal during casting. The main fracture seems, therefore, to be the result of bifilms. A generous quantity of bifilms is predictable in many early gravity-poured turbine blade castings. Those relatively few grains which revealed striations show the action of a fatigue failure mode for those regions of the metal where bifilms happened to be absent.

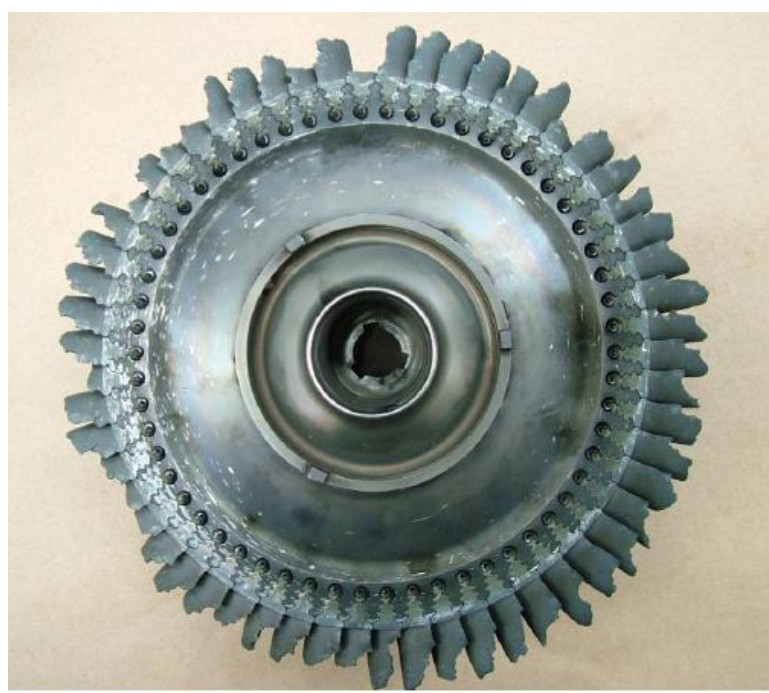

**Figure 5.** The extensive damage to the turbine by the catastrophic fatigue failure of one blade during a flight [39].

The identification of the conchoidal areas as due to bifilms, confirming bifilms to be the major failure mechanism, is corroborated below.

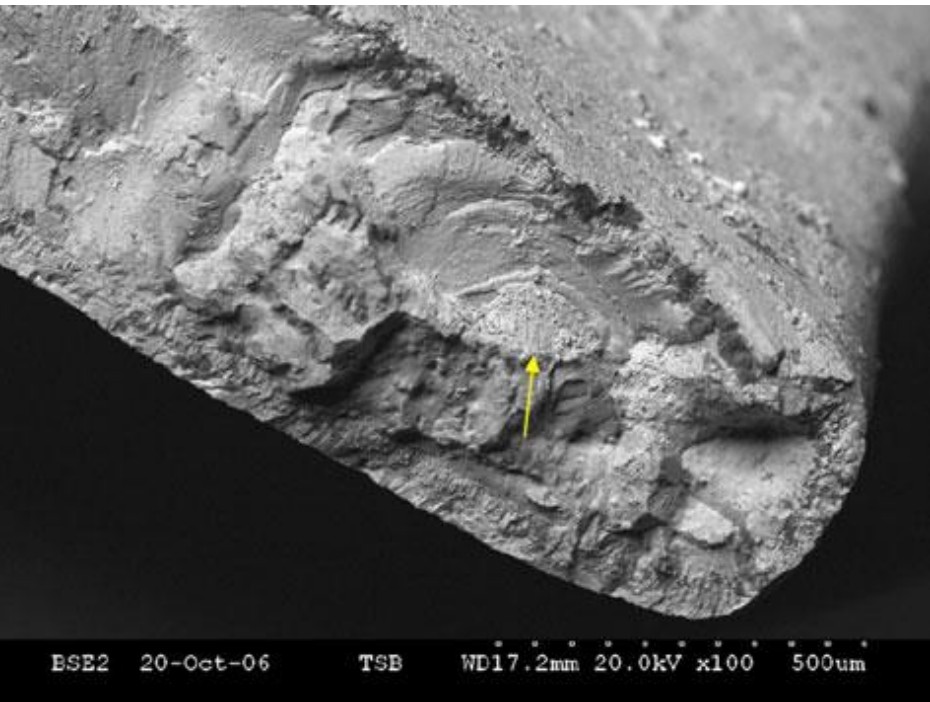

**Figure 6.** Failed turbine blade. Many grains had the appearance of conchoidal fracture surfaces; difficult to explain unless from the billowing shapes of bifilms. Very few areas of striations could be found on the fracture surface, suggesting a limited amount of fatigue in this so-called fatigue failure [39].

Figure 7 shows a section through the fracture surface. It reveals an array of oxide double films, i.e., bifilms, on a large grain (revealed by etching). Interestingly, they have a feature which is a defining characteristic of bifilms: there are three which are parallel, probably following a crystal plane (as illustrated schematically in Figure 3). The growth of grains and dendrites organizes the bifilms in this way, so it is to be expected that multiple bifilms can sometimes be parallel in a single grain. This is not the case for a crack grown by stress, in which the first crack effectively relieves the stress in its immediate surroundings, inhibiting the growth of additional parallel cracks.

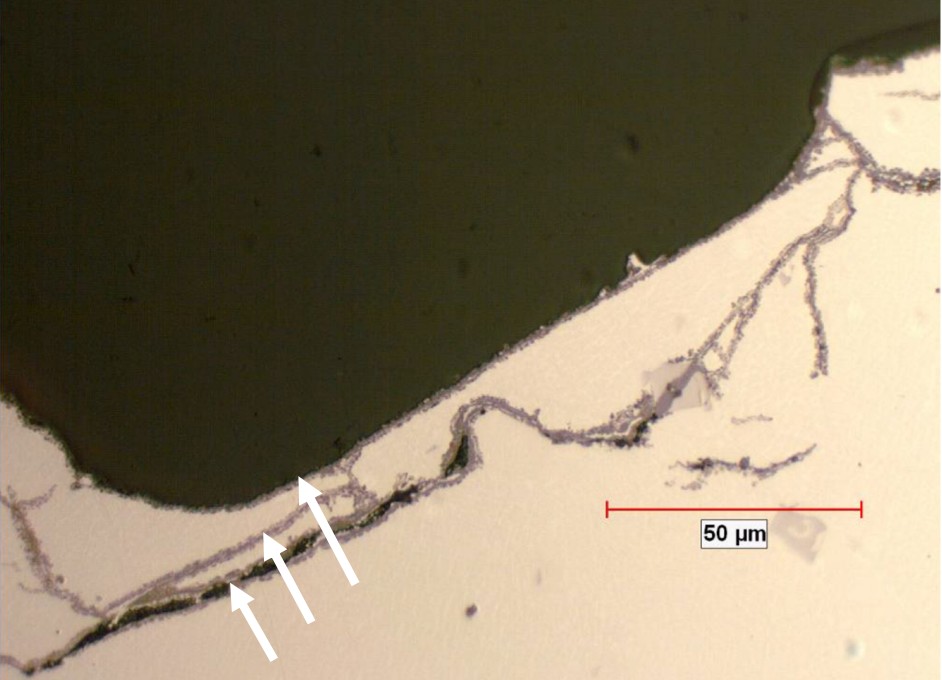

**Figure 7.** A section through the fracture surface, the main fracture displaying the remaining half of the double oxide film which formed the main pre-crack. Two additional oxidised film-like features, one of which is clearly a double film, form a close group of three parallel cracks (arrowed) [39].

The bifilms identified in the metallographic sections clearly occupy between 50 and 90% of the width of the blade. This leaves relatively little of the area to be fatigued, confirming the paucity of 'real' fatigue evidence on the fracture surface.

The reader might ask, 'Why did this blade fail by fatigue, while dozens of neighbors displayed no signs of any problem, many even surviving even after being struck by fragments of the failed blade?'. This is the wildly stochastic nature of bifilms: they are produced in the turbulence of the liquid surface, and the nature of turbulence is chaos; the situation is never precisely reproducible. Liquid turbulence has traditionally introduced a fatal lack of control in metallurgical processing, which has often proven to be baffling.

A brief search online [40–43] reveals numerous examples of failed equiaxed crystal turbine blades which appear explainable by bifilms. Examples can be given where the bifilm is suggested not only by the presence of its oxide, but also its favoured precipitates of carbides and topologically close packed (TCP) phases [10,44].

The number of failures of single crystal blades is much less, and almost certainly indicates a real benefit of single crystal blades as suffering fewer fatigue failures. This is predictable, because, for polycrystalline blades, the grains grow quickly from all directions, pushing ahead bifilms which are floating, suspended in the liquid, therefore trapping bifilms between converging grains. In contrast, the single crystal growth is unidirectionally upwards, sufficiently slowly to allow the floating ahead of bifilms, and those which do not float can be pushed, easing the bifilms out of the crystal and into the top feeder of the

casting, leaving the main part of the casting relatively free from bifilms. It is this freedom from bifilms which probably explains its excellence in creep. The traditional transverse grain boundary argument is clearly not correct as a result of grain boundaries now being proven, by molecular dynamics (MD) studies, to be generally nearly as strong as the matrix [45,46]. Incidentally, equiaxed castings could probably achieve similar properties to a single crystal castings if free from bifilms, thereby offering massive potential economy for turbines of all types.

*5.2. Case 2: Fatigue Failure of Steel Helicopter Components*

In 2012, two Super Puma Eurocopters ditched in the North Sea as a result of a fatigue failure [47]. Fortunately, all occupants survived. However, in 2016, a Norwegian Eurocopter lost its rotor and crashed killing all 13 occupants [48]. In all three cases fatigue had apparently initiated from a corrosion pit. The accident investigation teams were baffled, because in each case the corrosion pit appeared insignificant, far too small to have initiated fatigue, particularly in view of the huge safety factors (sometimes as high as a factor of 5) often used for such vital components.

Currently accepted thinking on the initiation of fatigue from etch pits involves such features as internal grain boundary oxidation (curiously suggestive of a bifilm) and the mechanism of stress corrosion cracking, requiring chloride-containing corrodent. However, failures are not confined to marine applications. Worse still, in 2018, following the tragic loss of a Korean helicopter by fatigue of its main drive shaft, several spare drive shafts held in store were taken out and tested. They were all found to contain severe cracks. The spare shafts had been provided from a respected source, Airbus Europe. Furthermore, of course, the shafts had not been in service and, therefore, not subjected to either stress or corrosion [49]. Clearly, the major cracks in helicopter drive shafts pre-existed in what had been thought to be sound metal.

Moreover, it is significant that these components were manufactured from vacuum arc remelted (VAR) steel. There has always been direct evidence that bifilm defects exist in VAR ingots because, as cast and unpeeled, VAR ingots simply crack when forged. This behaviour contrasts with otherwise identical ingots manufactured by electroslag remelting (ESR) which forge like butter. It is only recently that the fundamental defects of the VAR process have been understood and are now predictable. In the meantime, we have become complacent, accepting the faulty process but relying on non-destructive testing, unfortunately working at, or beyond, its limits of detection for many bifilms, especially the horizontal bifilms in VAR ingots; these oxide cracks, formed in 'vacuum', are probably only nanometres thick.

The VAR process manufactures bifilms by the unusual mode of advance of the melted metal, its meniscus rolling over the areas of frozen metal in contact with the water-cooled walls. The frozen metal at the side of the central pool oxidizes in the poor vacuum, developing an extensive horizontal oxide film. The advancing meniscus rolls out over the film, laying down its own oxide film onto the substrate film, forming a large horizontal bifilm. There is evidence that the bifilms may be up to 50 mm deep from the ingot surface, and so hardly influenced by 'peeling' (the operation to remove approximately 5 mm of the ingot surface).

The observation of the apparent initiation of fatigue failure from etch pits requires further elucidation. As suggested in 2020 [10,50], it seems probable that etch pits form where bifilms intersect the exterior surface of a metal, because the bifilm provides a route for corrodents into its interior. This process is enhanced by the precipitation of second phases on the outer surfaces of bifilms. The volume and shape changes of the new phase are more easily accommodated by the 'air gap' of the bifilm. There seems little doubt that inclusions do not precipitate on grain boundaries but precipitate on bifilms, since the saving of the energy of plastic work is of the order of 100 times higher for bifilms, compared to the saving of interfacial energy for grain boundaries. The plastic opening of bifilms in the solid state by the precipitation of inclusions aids ingress for corrodents, and the inclusions on the

exterior interfaces of the bifilms provide electrochemical couples to speed corrosion. The first outcome of these processes will be the creation of an etch pit (Figure 8). It is reasonable to conclude, therefore, that the existence of an etch pit is the witness to the pre-existence of a bifilm at that location. Furthermore, of course, the bifilm will be expected to be vastly more extensive than the etch pit.

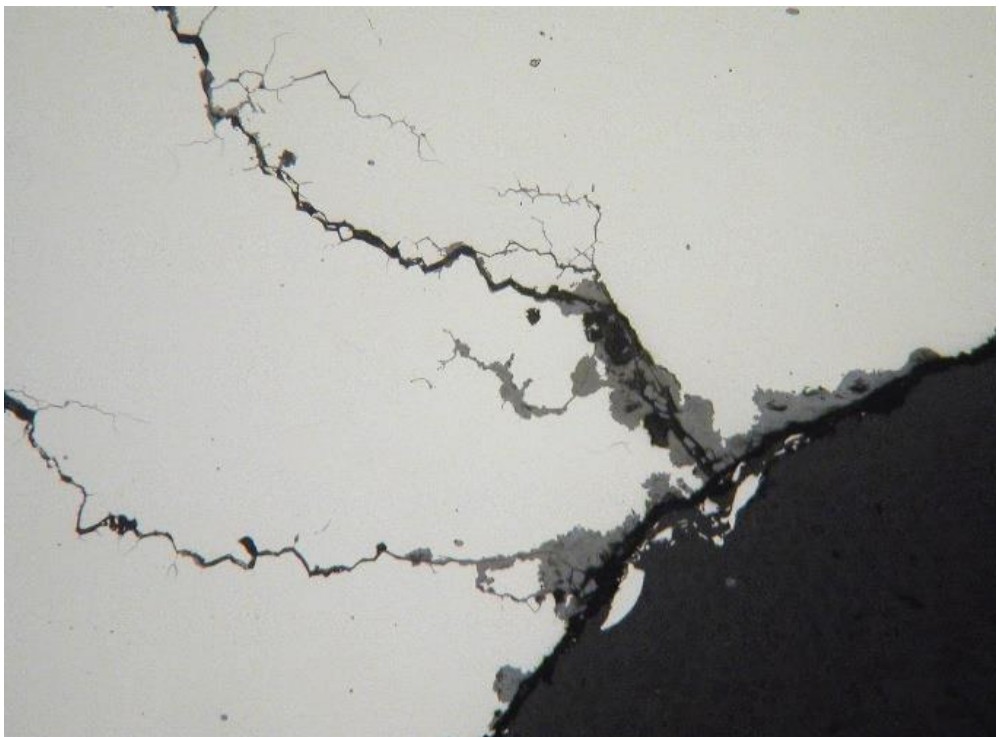

**Figure 8.** A detail of a large hemispherical etch pit, initiated by the two pre-existing bifilm cracks (rather than the pit initiating the cracks), in a steel blade of a steam turbine Reprinted with permission from ref. [51]. Copyright 2022 Elsevier.

Moving on to another example of a fatigue failure in a helicopter, details of a steel conformal gear is presented in Figure 9 [52]. The conformal gear in the mechanism is shown in Figure 9a, and the failed gear is presented in Figure 9b. The analysis of the fracture surface showed that fatigue initiated from a subsurface defect in the boxed area shown with an arrow in Figure 9c. X-ray map of the area showed the presence of oxygen (Figure 9d) and aluminium at that spot, leaving no doubt that an alumina ($Al_2O_3$) bifilm caused the in-service failure.

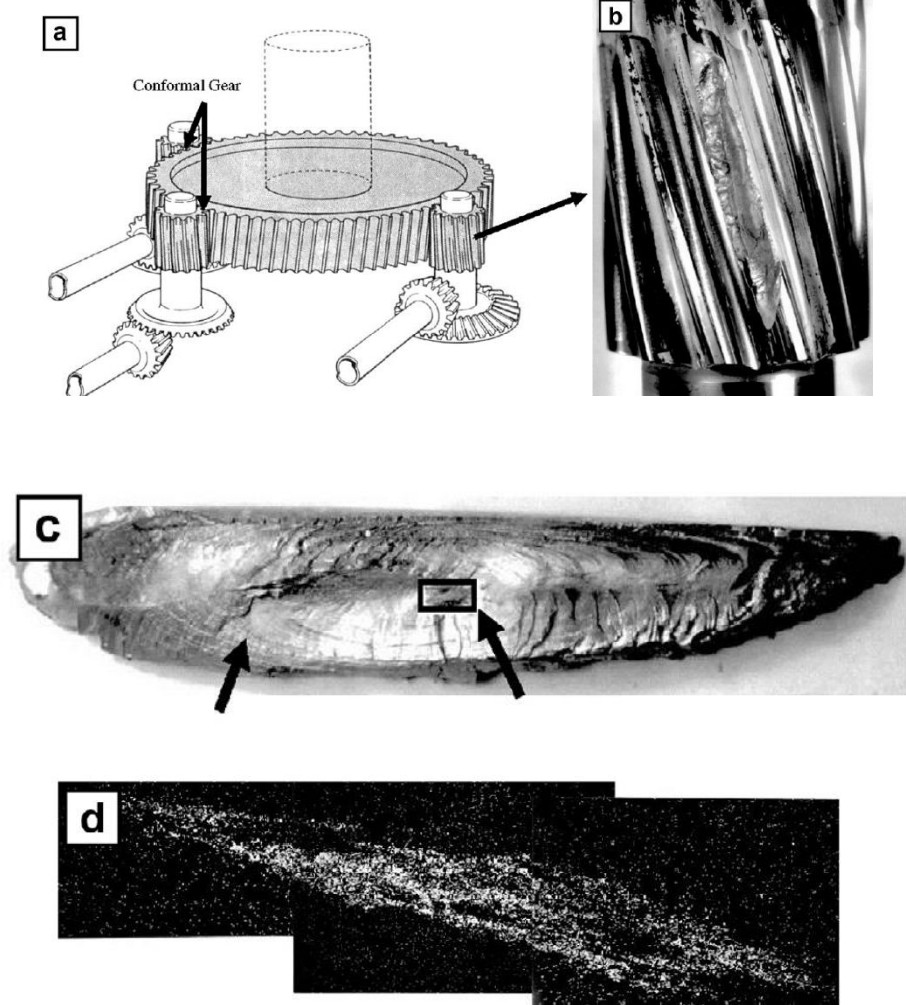

**Figure 9.** The details of an in-service failure of a steel helicopter conformal gear: (**a**) the gear assembly, (**b**) the gear that failed due to fatigue, (**c**) analysis of the fracture surface, showing the fatigue crack initiation site (boxed) and striations, and (**d**) X-ray map showing the presence of oxygen at the crack initiation site Reprinted with permission from ref. [52]. Copyright 2022 Elsevier.

*5.3. Case 3: Fatigue Failure of Magnesium Helicopter Housings*

Magnesium castings used in helicopters as gearbox housings seem to have suffered from fatigue failure due to entrained surface films. These are generally large castings, generally moulded in sand and cast in air. The traditional filling system designs are now known to be poor, so that oxide film entrainment to form bifilm cracks is likely. Davies et al. [52] analysed the fatigue crack initiators in helicopter magnesium gearbox casings, and reported that more than 70% of the failures can be attributed to bifilms (and pores). An example of a cast WE43 magnesium alloy gearbox housing that failed in-service due to fatigue is presented in Figure 10, which shows a large oxide film occupying a large area of several square millimeters that served as the main fatigue initiator (although a second bifilm crack can be seen, which also contributed to fracture).

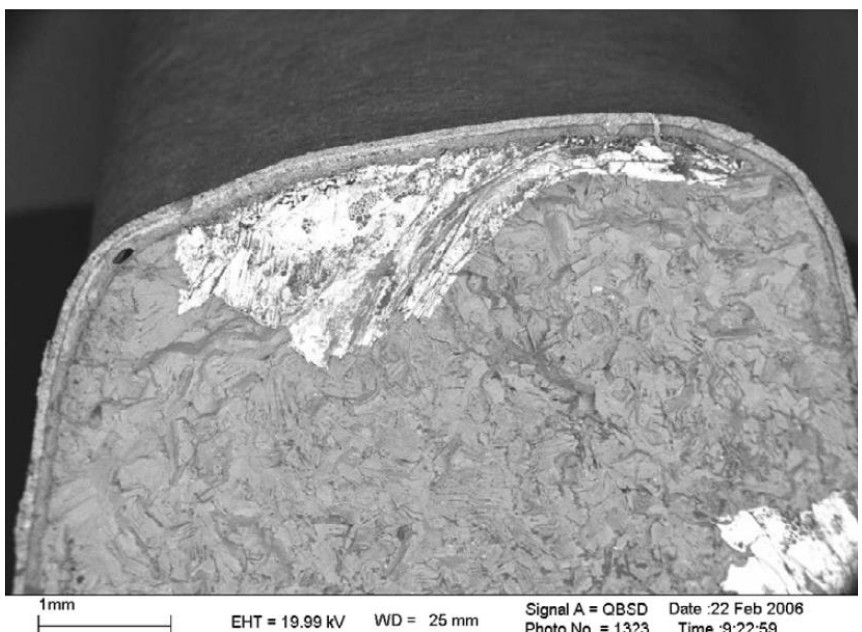

**Figure 10.** An oxide film exposed on the fracture surface of a WE43 magnesium alloy casting Reprinted with permission from ref. [52]. Copyright 2022 Elsevier.

In a later analysis of helicopter housings, Belben [53] showed that almost all fatigue initiators of the housing that failed in-service are attributed to entrained surface oxides. Such an oxide was found near the cracked web of a cast WE43A magnesium alloy casting in Figure 11.

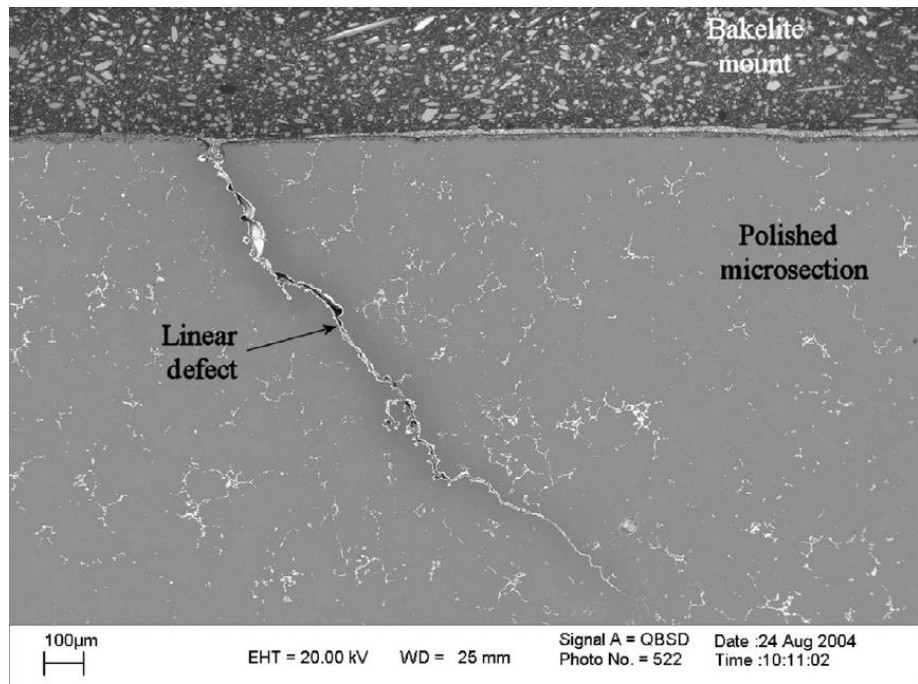

**Figure 11.** A bifilm crack over 1 mm long found adjacent to the cracked web on a cast Mg alloy housing Reprinted with permission from ref. [53]. Copyright 2022 Elsevier.

*5.4. Case 4: Fatigue Failure of the Cylinder Head of an Airplane Engine*

Airplanes that are powered by piston engines make up a large majority of the fixed wing fleet. In these airplanes, one of the leading causes of power loss and accidents is the

separation of cylinder heads in service and, therefore, loss of power during flight. Such failure was reported [54] in two cases of piston engine failure in an Utva-75 aircraft that occurred within a period of four months of each other. In both cases, in-service failure of their cast aluminium alloy cylinder heads, and consequent loss of power during flight, was attributed to fatigue. Failure analysis of the regions adjacent to the region of the fatigue fracture surface is presented in Figure 12. Metallographic analysis near the fracture surface [54,55] showed a high density of bifilms. Such damage given to liquid metal during melt preparation and casting results in significant weakening of the metal that subsequently results in premature, (potentially) catastrophic, failure due to fatigue.

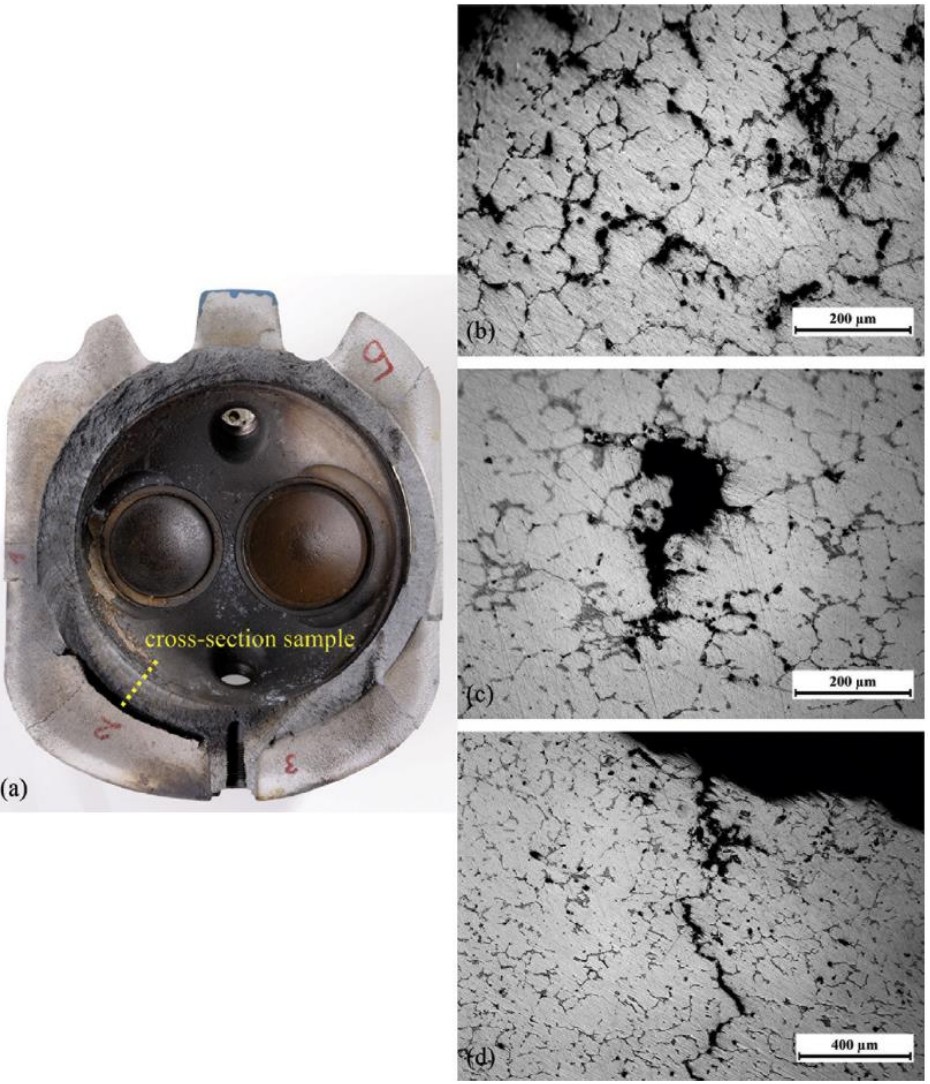

**Figure 12.** The failure analysis of the cylinder head that failed in service: (**a**) the failed cylinder head, (**b**–**d**) bifilms that were found in the area adjacent to the failed region Reprinted with permission from ref. [54]. Copyright 2022 Elsevier.

### 5.5. Case 5: Fatigue Failure in Aluminium Airplane Frame Castings

Li et al. [56] reported the in-service failure of an airplane frame component casting of an Al–Cu–Mn–Ti alloy. Two failed castings are shown in Figure 13. Note that pores are visible in Figure 13b. As discussed previously, pores are only the visible part of the damage given to liquid metal, and, therefore, actual entrainment damage is larger (as is evident in Figure 14). Moreover, electron microscopy of the pores on the fracture surface showed extensive oxide films, as shown in Figure 14. It is only realistic to expect that bifilms extend

into the matrix and remain as unopened at the end of solidification. However, in a tensile stress state, they open right away, concentrating the tensile load into the remaining area of the frame, leading to the almost immediate formation of a propagating fatigue crack. This is shown in the results of Li and Hu [57], who investigated why an Al-7%Si-Mg alloy casting for an airplane frame failed in vibration tests. They found a large number of pores, as well as bifilms associated with the pores, that extended into the aluminium matrix, as shown in Figure 15. These features, also referred to as fissures, demonstrate the extensive damage given to the liquid metal prior to solidification. We can also see that a densification process, such as hot isostatic pressing, would close this pore but would not heal the damage, at least not completely, as was observed in A206 castings [8,9].

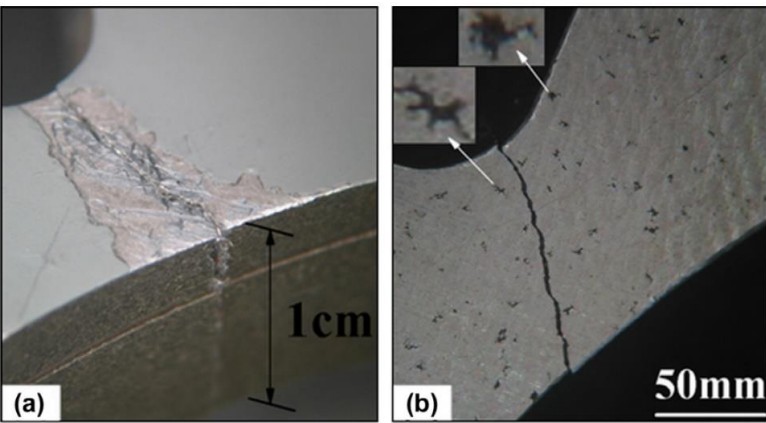

**Figure 13.** Two failed airplane frame castings (**a**,**b**) of an Al–Cu–Mn–Ti alloy. Pores are visible in (**b**) Reprinted with permission from ref. [56]. Copyright 2022 Elsevier.

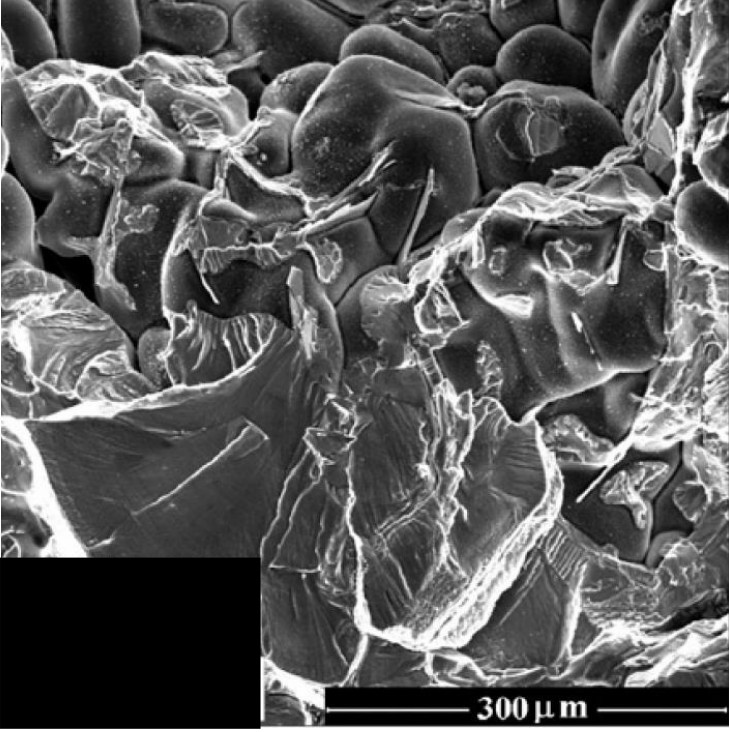

**Figure 14.** A pore found on the fracture surface of one of the castings in Figure 13, showing the presence of oxide films on the surface of the pores. The films are the remaining half of the bifilm, or bubble, and appear to have been sucked into the inter-dendritic regions, probably as a result of shrinkage during freezing Reprinted with permission from ref. [56]. Copyright 2022 Elsevier.

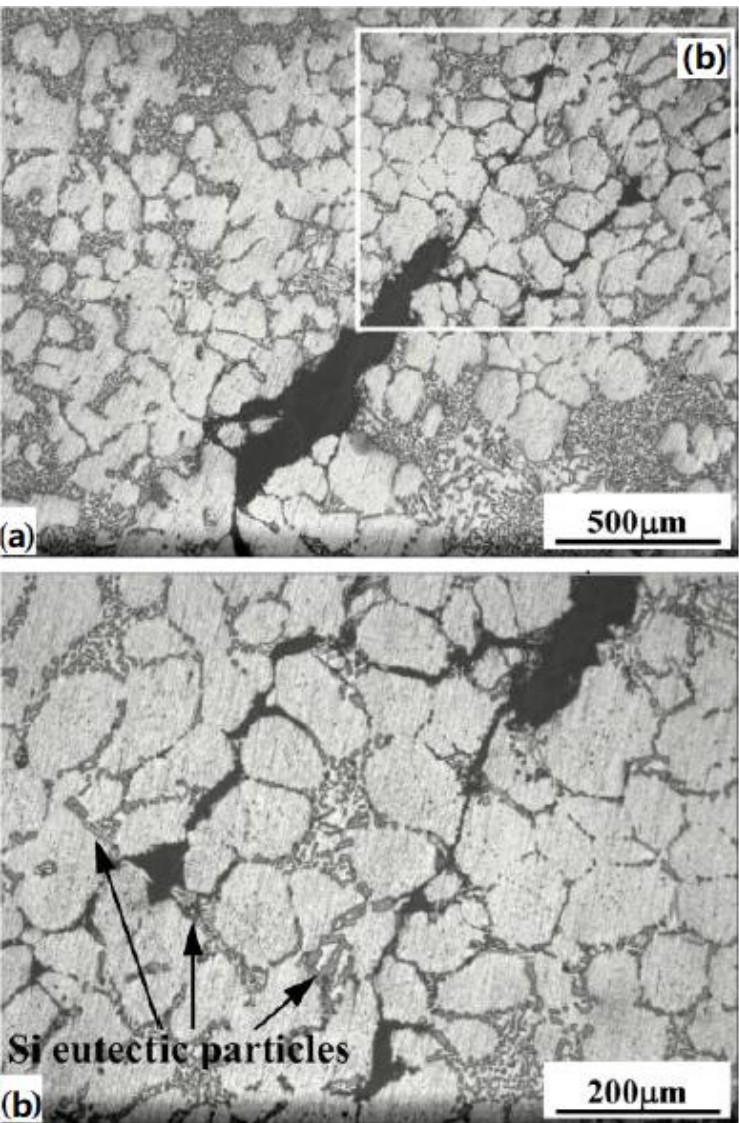

**Figure 15.** Bifilm (part of which is opened as a pore, while other parts remain closed as cracks) extending into the aluminium matrix found near the failed region of an Al-7%Si-Mg airframe casting. (**a**) the overall defect, (**b**) the magnified version of the square in (**a**) showing the unopened bifilms associated with the pore. Reprinted with permission from ref. [57]. Copyright 2022 Elsevier.

## 6. Conclusions

1.  Naturally, although instances can be quoted in which a failure can be 100% attributed to the progress of a fatigue crack, this is far from the examples presented in this report and may be limited to the fatigue of laboratory test pieces.

2.  Studies of the fatigue of laboratory test pieces will always be of interest, but will be of little value to the lifetime predictions of engineering components if the engineering components contain the currently normal populations of bifilm defects from the casting process.

3.  In the instances quoted in this report, which appear to be typical of a large proportion of fatigue failures of components of rotating machinery, the overwhelming contributors to the failure are bifilms generated in the inappropriate casting technique despite appropriate casting techniques now being available [22,44]. The fatigue contribution to failure is relegated to the small regions of the component crack path which happen to be absent of bifilms. A prior study [38] found bifilm defects to be responsible for up to 99% of the 'fatigue' failure of a wind turbine main bearing. From this study, the

bifilm contribution to the 'fatigue' failure of the turbine appeared to be somewhere in the region of 50 to 90%. The bifilm contribution for the 'fatigue' failure of the helicopter drive shaft cannot be discerned from the accident investigation report, but would be expected to be in the region of at least 10 to 50%.

4. It seems bifilm cracks and fatigue processes are both necessary for complete failure by what we generally call 'a fatigue failure'. It appears probable that elimination of bifilms would eliminate most in-service fatigue failures.

**Author Contributions:** The authors collaborated in all aspects of this manuscript. All authors have read and agreed to the published version of the manuscript.

**Funding:** This work received no external funding.

**Institutional Review Board Statement:** Not applicable.

**Informed Consent Statement:** Not applicable.

**Data Availability Statement:** Not applicable.

**Conflicts of Interest:** The authors declare no conflict of interest.

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
