# Peer review of "Fatigue Failure in Engineered Components and How It Can Be Eliminated: Case Studies on the Influence of Bifilms"

_metals, doi:10.3390/met12081320_

Round 1

Reviewer 1 Report

 This study illustrates five examples in which pre-existing bifilm defects  act as crack initiator. The bifilm defect occurs naturally when a metal is poured. Until recently, the ubiquitous presence of these very thin double films has not been widely accepted because it is difficult to detect these thin but extensive defects.

5 case Studies on In-Service Fatigue Failure due to Bifilms are presented:

 a turbine blade,  a steel helicopter components, a magnesium helicopter housings, a Cylinder Head of an Airplane Engine and  an Aluminium Airplane Frame Castings.

 Each case are well documented. The paper is easy to read in correct English and the quality of pictures is high.

 This paper is accepted as received.

Author Response

We would like to thank this reviewer.

Reviewer 2 Report

The article reports interesting case studies of components failured due to fatigue, underlining how these failures are due to the presence of bifilm oxides resulting from bad foundry practice. These bifilms also persist in subsequent stages of thermo-mechanical processing and according to the authors will cause future failure of the components. There is nothing to complain about this, just as there is nothing to complain about the fact that it is necessary to improve the foundry practice to avoid the formation of bifilms.

According to the authors, using classical fatigue tests can be misleading to predict the fatigue behavior of components. This, however, is not clear to me.

Two basic methodologies are commonly used to address the problem of fatigue in materials. The first of these methodologies is the traditional fatigue life (or S/N) analysis, where no notch, nominally free of defects, samples are cycled at various levels of stress (S) or strain to determine the number of cycles at break (N). In this approach the fatigue life concerns both the nucleation and the propagation of the crack up to the final failure. In this case the presence of defects is essential to define the fatigue limit of materials. For instance, spheroidal cast irons produced in thin sections, with few defects, have a much higher fatigue limit than cast irons with the same chemical composition, but produced in heavy sections. The second is a damage tolerance approach in which the fatigue crack propagation is characterized by the crack growth rate per cycle, defined by da/dN, as a function of the linear-elastic stress intensity range, defined as DK = Kmax - Kmin, where Kmax and Kmin are the maximum and minimum stress intensities during the load cycle. Maybe the authors referred to the second approach only? However, usually in an appropriate study both approaches are used, and with the Kitagawa-Takahashi diagram the characteristic dimension of the defect is found for materials. Furthermore, there is another possible problem. If the test samples for the study of fatigue behavior come from a sacrificial component that has followed all the various stages of production, it will also have all the typical defects of a component put into operation. The statistical study of the fatigue behavior of the material is representative of what the probability of the material/component breaking will be, if there is consistency between the material of the component and the material of the fatigue samples. It is different if the material is taken from a cast block with similar cooling conditions of the as-cast component. In this case the quality of the material could be better and therefore the results of lab investigations might be misleading. International standards help manufacturer and purchaser to properly define fatigue sample pickup from as-cast components. If the material is selected from the most critical point, the statistical study is useful for the purpose of design. If the selection is wrong, the statistical study would no longer be useful, assuming that some unexpected flaw could arise anyway. However, the study of fatigue behavior is statistical, and there may always be some defect that is over the characteristic dimension. The better the foundry practice, the lower the probability of having defects, even catastrophic ones that can lead to failures outside of the predictions coming from the statistical study of the fatigue behavior of the material. While I think the article is interesting, I have not grasped the innovative side that should be the heart of a scientific article. It hasn't been explained well, at least according to me, in what sense the lab results are misleading. It is necessary to enter into the merits of this statement, explaining in details what the authors mean by the statement that the fatigue properties obtained in the laboratory can be misleading.

I believe that the article should not be published, at least in the form of a scientific article, and it has to be improved.

Author Response

We sincerely appreciate the detailed comments of the reviewer.  As a response to the comments, we have added a section on the effect of bifilms on fatigue crack initiation and propagation based on in situ studies.  In this section, we have also outlined our views on why fatigue studies have had incomplete success (please note the change in the reference to earlier fatigue studies).  As a result, we think that the paper is much improved now.

Reviewer 3 Report

The authors begin with a flawed thesis and then attempt to support it by presenting 5 case studies that are misconstrued. It is actually very disparaging to suggest that after a century of investigation and scientific and engineering advances that metal fatigue is not understood and that engineering does not prevent fatigue failures. Failure does occur, but it is generally due to circumstances, some which is within our control and some beyond; the fact that billions of components do not suffer fatigue failure is brushed aside in favor of the few cases where inspection protocols were not followed, or inherent material variability and lack of quality control during manufacture get the better of the fatigue process.

Author Response

We thank the reviewer for the comments. We have struggled to understand why our thesis is flawed because the reviewer has not provided any feedback on this. Moreover we have strengthened our argument by using recent observations from in situ fatigue tests. We think that the paper is now much improved.

Also based on this reviewer\s comments, we decided to revise our comments and refer to the past efforts in fatigue literature as "incomplete success".  We thank the reviewer for emphasizing this fact.

Reviewer 4 Report

Very nice piece of work which will be of very high interest for the readers. No modifications or changes to be done. Should be published asap as there may be direct interest of part of the audience for similar problems of high practical interest.

Author Response

We thank this reviewer for the comments.

Round 2

Reviewer 2 Report

The authors added some text to clarifying successfully the comments from the reviewer.

 There are some repetitions in the text, for instance:

Line 160:    ... to open to expand....

Line 219:   ... of some of some...

Author Response

Corrections are made. We thank the reviewer.

Reviewer 3 Report

The reviewer appreciated the authors’ acknowledgement. Their thesis is flawed because they cite a few cases where, for reasons beyond the decades (centuries) of research and development in understanding fatigue design, implementation, inspection and maintenance - in service failures have occurred. Our understanding of fatigue processes, and the subsequent development of design standards, has been an unqualified success in ensuring safety. There are miriad circumstances in industry that lead to failures, but in the current state of the art it is unlikely to be a misunderstanding of the processes, but rather a lack of proper maintenance and/or inspection that have lead to in service failures.

Author Response

We appreciate the reviewer's comments and time. However, we disagree completely with her/his stance.  We also understand that our differences cannot be reconciled at this point.  We do plan to present new research findings supporting our views.